

# Atmospheric Rivers as Triggers of Compound Flooding: Quantifying Extreme Joint Events in Western North America Under Climate Change

Andrew Vincent Grgas-Svirac[1], Mohammad Fereshtehpour[1], M. Reza Najafi[1], Alex J. Cannon[2],
Hamidreza Shirkhani[3]

[1]Department of Civil and Environmental Engineering, University of Western Ontario, London, Canada.
[2]Climate Research Division, Environment and Climate Change Canada, Victoria, BC, Canada.
[3]National Research Council Canada, Ottawa, ON, Canada

Corresponding author: M.R. Najafi (mnajafi7@uwo.ca)

**Abstract.** Atmospheric Rivers (ARs) are narrow bands of concentrated moisture that transport water vapor from the tropics to higher latitudes. They are responsible for ~90% of poleward water vapor transport and play a vital role in water resource management along the North American west coast. While ARs significantly contribute to regional water supplies, they are also major drivers of flooding. This study investigates the extent to which ARs contribute to compound inland flooding (CIF) events where multiple drivers intensify flood risks, namely Rain on Snow (ROS) and Saturation Excess Flooding (SEF) events. Furthermore, the influence of internal climate variability is investigated relative to anthropogenic climate change. Using the CanRCM4 Large Ensemble simulations, we analyze the frequency and seasonality of AR-driven CIF events in Western North American coastal areas, with a focus on understanding how ARs interact with additional factors such as snowpack and soil moisture. ARs are shown to be dominant drivers of CIF events by contributing to the development and intensification of these events. These conditions also shape the seasonality and intensity of AR-driven CIFs. Projections suggest that internal climate variability can significantly contribute to future uncertainty in CIF frequency and intensity, complicating efforts to predict and mitigate these events. The findings underscore the importance of integrating AR-related flooding risks into flood management strategies and infrastructure design to adapt to a changing climate.

## 1. Introduction

Atmospheric Rivers (ARs) are synoptic-scale phenomena characterized by long, narrow bands of concentrated water vapor that transport moisture from tropical and subtropical regions to higher latitudes. They frequently form in association with



extratropical cyclones and are a key driver of heavy precipitation and flooding in mid latitude regions (Ralph et al., 2017, 2020). Despite only occupying about 10% of space on the globe at a given time, they are estimated to be responsible for 90% poleward water vapor transport across the mid latitudes (Ralph et al., 2020; Zhu & Newell, 1998). Along coastal regions in the mid latitudes, ARs play an important role in local water resource management, often serving as primary contributors to

annual precipitation. This is particularly evident during the autumn and winter seasons along the North American west coast, where ARs deliver the majority of precipitation and significantly influence regional hydrology (Lamjiri et al., 2018; Rutz et al., 2014). In the State of California, the majority of annual precipitation has been found to be associated with a few very strong AR related winter storms (M. Dettinger, 2016; M. D. Dettinger et al., 2011). AR related precipitation contributes significantly to both rainfall and snowfall. At higher elevations or in colder regions, they are key providers of snow pack that can act as

natural reservoirs to support seasonal water resource availability for agricultural, ecological, and urban needs (Guan et al., 2010, 2016; Rutz et al., 2014). Studies in British Columbia indicate that ARs contribute approximately 90% of annual extreme precipitation, and account for more than 33% of annual total precipitation (Sharma & Déry, 2020a). Similar findings have been reported for landfalling ARs in Southwest Alaska, where ARs also play a dominant role in extreme precipitation events (Nash et al., 2024; Sharma & Déry, 2020a).

Landfalling AR storms deliver substantial precipitation that often provide beneficial impacts, such as drought relief and reservoir replenishment. In California alone, 74% of droughts have ended upon the arrival of an AR (Dettinger, 2013). However, the intense precipitation brought by ARs also act as a driving force of flooding events. In the western United States, ARs are one of the primary causes of flooding. For example, in California's Russian River Basin all major floods between 1997 and 2006 have been attributed to landfalling ARs (Ralph et al., 2006). Comprehensive studies of the entire U.S. west

coast have shown that a large portion of annual runoff peaks are associated with AR occurrence (Barth et al., 2017; Konrad & Dettinger, 2017). In British Columbia's coastal region, research indicates that for many watersheds, up to half of annual streamflow is linked to landfalling ARs (Sharma & Déry, 2020b). Furthermore, one of the most severe flooding events in the province's history was when two ARs made landfall in November 2021. The result was widespread infrastructure failure due to flooding and landslides; with damages totaling 65 million CAD in damages, the displacement of 20 000 residents and five

deaths (Gillett et al., 2022; Richards-Thomas et al., 2024). Most recently, in October 2024, a strong AR made landfall in BC, delivering 150–300 mm of rain, which overwhelmed metropolitan storm sewers resulting in widespread flooding (Cassidy, 2024). Overall, substantial evidence links severe flooding in North America's west coastal regions to landfalling ARs.

Given their ability to deliver large amounts of precipitation rapidly, ARs often amplify the severity of flooding events when combined with other contributing factors. This makes ARs a significant driver of compound inland flooding (CIF) events,

where multiple mechanisms interact to produce extreme runoff. A compound event is defined as an event with amplified severity due to the contribution of multiple drivers or hazards (Hao & Singh, 2020; IPCC, 2012; Zscheischler et al., 2020). Examples of compound flooding events include Rain on Snow (ROS) events and Saturation Excess Flooding events (SEF). An ROS event occurs when intense rainfall falls on snowpack generating large amounts of snowmelt, amplifying runoff generation. SEF events occur when soil is saturated prior to an intense rainfall event, leading to an amplified runoff response



due to the lack of infiltration. Recent high-profile catastrophic AR-related flooding events indicate that although ARs often act as primary drivers of extreme precipitation, the resultant runoff response is frequently amplified by pre-existing conditions such as saturated soils or snowmelt. For example, the extreme runoff generated in February 2017, which led to the failure of the Oroville Dam spillway, was driven by a combination of unusually deep antecedent snowpack undergoing rapid melting, high antecedent soil moisture and the arrival of an AR storm (Henn et al., 2020). Similarly, during the November 2021 British

Columbia floods ARs were identified as the primary driver, however, a large portion of the impact was also attributed to snowmelt which amplified the runoff response (Gillett et al., 2022). These flooding events involved multiple contributing factors, including AR-driven precipitation combined with antecedent conditions such as snowmelt and saturated soils emphasizing the complex interactions that often modulate CIF events. However, most AR-related studies focus primarily on the extreme precipitation or runoff generated by individual landfalling AR events and often overlook the compounded effects

of multiple drivers. This gap in understanding can lead to an underestimation of the overall hazard (Singh et al., 2021; Wazneh et al., 2020). The importance of considering the dependence between multiple drivers in relation to AR events is underscored by previous studies, which show that even lower-intensity ARs can generate significant runoff when combined with factors such as saturated soils or snowmelt (Chen et al., 2019; Konrad & Dettinger, 2017).

In a warming climate, AR events are generally projected to become more frequent and intense (Gonzales et al., 2019; O'Brien

et al., 2022; Radić et al., 2015; Shields et al., 2023), which is expected to exacerbate flooding conditions in regions that regularly experience AR-driven events. In the previously discussed regions, several studies have already explored the potential impact of more frequent and intensified AR events under future warming scenarios. Current evidence suggests that projected changes in AR characteristics are likely to result in more frequent and severe flooding events along the North American West Coast. In California the projected changes in AR may result in large increases in AR-related extreme precipitation and the

elevated risk of flooding events (Gershunov, et al., 2019; Huang & Swain, 2022; Shields et al., 2018). Furthermore, projected changes in landfalling ARs are also associated with projected increases in fiscal flood losses relative to the historical period (Huang & Swain, 2022; Rhoades et al., 2021). In British Columbia, streamflow and extreme precipitation are projected to increase in proportion to the increase in AR related activity (Curry et al., 2019). Similar trends have been identified for AR activity in Alaska, with projections indicating corresponding increases in extreme precipitation and runoff (Nash et al., 2024).

Furthermore, several studies suggest that under a warming climate, the frequency of sequential AR events is expected to increase, further amplifying the future risk of AR-related flooding events (Bowers et al., 2023, 2024; Fish et al., 2022). Although AR storms are recognized as significant contributors to ROS and SEF events (Guan et al., 2016; Ralph et al., 2013; Rhoades et al., 2024; Shulgina et al., 2023), most existing AR studies primarily focus on extreme precipitation and its immediate impacts. While there has been some regional work examining AR interactions with ROS events and snow

accumulation (Guan et al., 2010; Kim et al., 2013), there is a critical gap in understanding the broader role of ARs in compound flooding events. This lack of research on the interaction between ARs and multiple drivers, such as SEF and ROS, may lead to an underestimation of their amplified role in future compound flooding, especially under a warmer climate where such interactions could become more frequent and impactful.



This study quantifies the contribution of ARs to compound inland flooding by investigating the co-occurrence probability of landfalling ARs, extreme precipitation and CIF events such as ROS or SEF along western North American coastal areas. Western North America has been chosen due to its historic vulnerability to AR related flooding events, as highlighted in previous studies. We also investigate the influence of seasonality on the frequency and intensity of these joint occurrences. Furthermore, recognizing the growing evidence linking an increase in AR occurrence to a warming climate, we evaluate the role of anthropogenic climate change and internal climate variability in AR-induced compound flooding under different warming levels. While previous research has primarily focused on ROS and AR on the U.S. side of the Pacific Northwest or specific case studies (Bowers et al., 2024; Guan et al., 2016; Rhoades et al., 2024), our study offers a novel perspective by examining how ARs contribute to both ROS and SEF as compound flooding mechanisms in a broader inland context. This dual focus allows us to assess the role of ARs in driving these mechanisms and to evaluate how their influence may shift under future climate scenarios. Additionally, this study builds on previous research by examining this phenomenon using the CanRCM4 model. Section 2 describes the datasets and methods employed. Section 3 presents the results, and Section 4 discusses the findings in the context of existing literature.

## 2 Methodology

### 2.1 Climate Data

The study employs the Canadian Regional Climate Model Large Ensemble (CanRCM4-LE), which is a high-resolution climate model that covers North America at a resolution of 0.44° x 0.44° (~ 50 km). CanRCM4 is driven by the second generation Canadian Earth System Model (CanESM2) large ensemble, using historical forcing from the Coupled Model Intercomparison Project Phase 5 (CMIP5) for 1950–2005 and Representative Concentration Pathway (RCP) 8.5 forcing from 2006 to 2100 (Scinocca et al., 2016; Singh et al., 2021). To effectively characterize compound events, a large sample of events is essential for a robust statistical analysis (Zscheischler and Seneviratne, 2017). This study utilizes 28 members from the CanRCM4-LE model to analyze compound events, ensuring a robust statistical assessment. The selection of 28 members aligns with previous studies, which have successfully characterized AR-related events using ensembles of similar size (Hagos et al., 2016; Michaelis et al., 2022; Tseng et al., 2022). Each ensemble member includes daily simulations of integrated vapor transport (IVT), precipitation, snowmelt, and soil moisture from 1950 to 2100. In this study, CanRCM4 is re-gridded to the 0.5° rectangular grid using bilinear interpolation and only land cells are considered in the following analysis (Figure 1). The data was re-gridded to facilitate future comparisons to bias corrected CanRCM4 datasets (Cannon et al., 2022).

The CanRCM4-LE dataset was generated by perturbing the initial conditions of the individual model, allowing for the characterization of internal climate variability. Since the differences between ensemble members arise solely from these initial condition perturbations, rather than variations in model structure, the approach isolates the effects of internal variability (Deser et al., 2020; Kay et al., 2015). To improve estimates of nonstationary extreme statistics, the large ensemble treats all simulations as approximately statistically independent, given the model's sensitivity to atmospheric initial conditions (Haugen et al., 2019;



Stein, 2020; Tebaldi et al., 2021). Previous studies have demonstrated that CanRCM4 performs well in reproducing key regional climate patterns relative to observed data (Il Jeong & Cannon, 2020; Whan & Zwiers, 2016). Furthermore, the model has been shown to represent AR activity with acceptable accuracy (Whan & Zwiers, 2016). To further validate the CanRCM4-LE dataset to represent AR activity, it will be evaluated against the ERA5 reanalysis dataset during the baseline period. ERA5,

developed and maintained by the European Centre for Medium-Range Weather Forecasts (ECMWF), is a global reanalysis dataset that provides a continuous and consistent record of atmospheric variables (Hersbach et al., 2020). ERA5 has been shown to reasonably capture AR activity (Collow et al., 2022) and is widely used in similar studies for AR analysis (Espinoza et al., 2018; Guan & Waliser, 2019; Lavers & Villarini, 2015; Rutz et al., 2014; Tseng et al., 2022). In terms of AR detection, ERA5 generally resembles observational satellite data with a slight positive (negative) bias in higher(lower) latitude regions

(Cobb et al., 2021; Ma et al., 2023). For comparison, hourly IVT data from ERA5 is aggregated into daily values and re-gridded to a $0.5° \times 0.5°$ rectangular grid to align with the resolution of the CanRCM4-LE dataset.

This study presents projections at four warming levels using 31-year time frames: the Baseline (BL) period (1986-2016) corresponding to a +1 ℃ global mean temperature change (GMTC) compared to the pre-industrial level (1850-1900), WL1.5 (2001-2031) corresponding to +1.5 ℃ GMTC, WL2 (2013-2043) corresponding to +2 ℃ GMTC, and WL4 (2053-2083)

corresponding to +4 ℃ GMTC (Cannon et al. 2020). For regional interpretation, the Bukovsky regions, as shown in Figure 1a is considered. These regions offer a simplified representation of North America's terrestrial ecoregions, serving as a useful proxy for identifying areas with similar climatological characteristics (Bukovsky, 2011). Figure 1a also shows the approximate location of dams within the region, providing insight into potential impacts on infrastructure. This data was obtained from the Global Dam Tracker database (Zhang & Gu, 2023). Figure 1b also depicts approximate elevation of the area as a point of

reference while discussing results since topography plays an important role in AR related events. Elevation data was processed from the Global Multi-resolution Terrain Elevation Data 2010(GMTED 2010) dataset created by USGS during the shuttle radar topography mission (Danielson & Gesch, 2011).



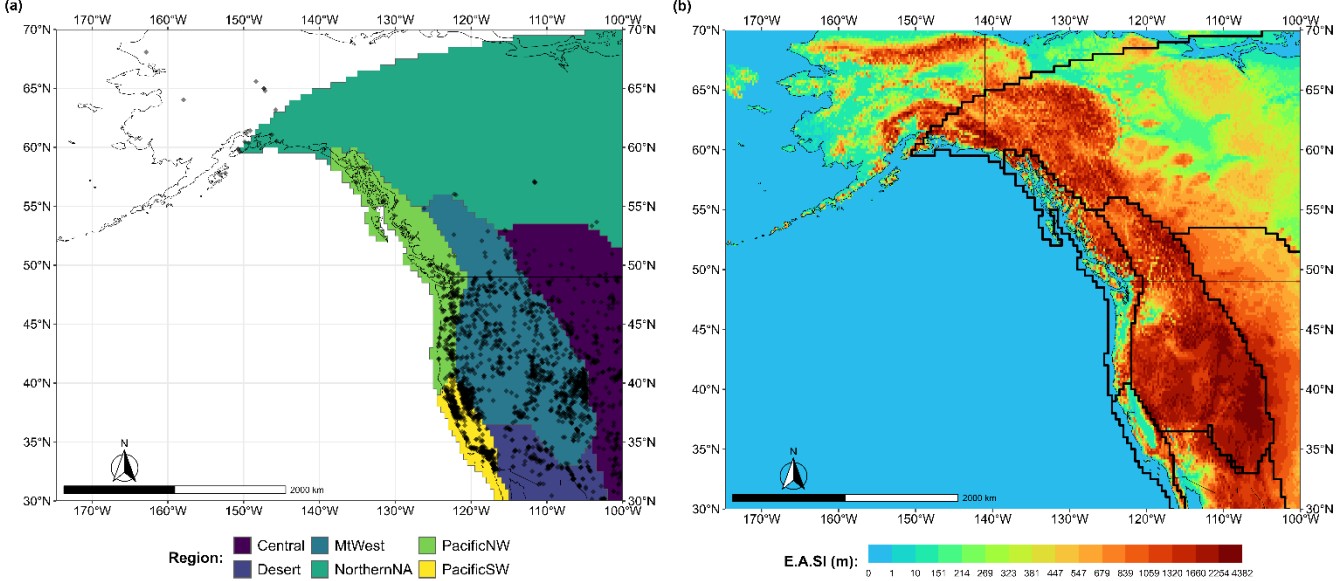

**Figure 1: Study area, a) subdivided by Bukovsky regions with the black diamonds indicating the location reservoirs from the Global Dam Tracker database. While b) depicts the approximate Elevation Above Sealevel (EASl)in meters at a resolution of 0.22°x0.22° from the GMTED 2010 dataset provided by USGS.**

## 2.2 AR Detection

ARs are identified using a Eulerian perspective developed by Ralph et al. (2019) that detects ARs using integrated vapor transport over time. A Eulerian perspective is a widely used approach that defines an AR as an observer would at a specific location monitoring conditions over time (Guan & Waliser, 2015; Rutz et al., 2014). The Eulerian perspective is selected for this study due to its compatibility with CIF event detection. In this study, AR conditions will be defined for a single cell in which IVT exceeds 250 kg/m/s for over 24 hours (1 day) according to the scaling provided by Ralph et al. (2019). A fixed IVT threshold of 250 kg/m/s is often considered at the minimal IVT threshold in AR studies for this region (Gershunov et al., 2017; Ralph, Rutz, et al., 2019; Ralph, Wilson, et al., 2019). Recognizing that weak to moderate ARs can have significant impacts, these thresholds align with ARs that are primarily category 1 or higher to allow for a wide range of AR events to be considered. For example, the AR associated with the 2021 British Columbia floods had a return period of just 10 years but the associated 2 day rainfall had a 50 to 100 year return period (Gillett et al., 2022). Since rainfall intensity is more dependent on total moisture convergence, most AR detection methods will not always fully represent the intensity of AR related storms, since they are detecting the AR based on the amount of moisture being transported (Mo et al., 2021).



### 2.3 CIF Event Definition and Likelihood

This study considers the likelihood of two different types of CIF events coinciding with a landfalling atmospheric river. Each type of CIF event is defined using a threshold definition of their respective drivers. The percentile-based approach takes advantage of the large amount of data provided by the large ensemble and avoids model biases (Poschlod et al., 2020). Each

quantile threshold is calculated based on the time series for each individual grid cell. The CIF events defined in this study are ROS events and SEF events. A high threshold is used to ensure the most severe events are identified. ROS events are identified when daily precipitation and snowmelt exceed the 98th percentile simultaneously within a single grid cell. In addition, snowmelt should contribute to at least 20% of the combined liquid generated during the event to ensure that ROS events detected in the CanRCM4 dataset are impactful, rather than instances where snowmelt is negligible. This threshold is a

conservative estimate for ROS events based on the approximate amount of snowmelt contribution identified in observational results (Freudiger et al., 2014; Musselman et al., 2018). SEF events are defined as days when both daily precipitation and soil moisture exceed their respective 98th percentiles. The likelihood of Extreme Precipitation (XPRA) Events coinciding with landfalling atmospheric rivers are also considered to further contextualize CIF likelihood. XPRA events are defined as days when 1-day precipitation exceeds the 98th percentile. For all identified extreme precipitation, ROS, and SEF events, surface

runoff must exceed its 98th percentile on the same day or the day immediately afterward. This additional criterion ensures that the events analyzed are strongly linked to significant runoff generation. It should be noted that a sensitivity analysis by Fereshtehpour et al. (2025) showed that incorporating event-based definitions with varying gap-day criteria (0–3 days) did not significantly alter the spatial patterns of CIF events, thereby supporting the robustness of the daily co-occurrence approach used in this study.

Events are defined as AR-related if they co-occurred with the pre-defined AR conditions described in Section 2.2. Once all CIF events are identified for the study area, the conditional probability of an event being AR-related is calculated as the total number of identified AR related events divided by the total number of events. This represents the likelihood of a CIF event being associated with an AR, given the occurrence of a CIF event (Equation 1). To assess the significance of these probabilities for future projections, a bootstrapping method is applied. A single process variable, in this case rainfall, is randomly reshuffled

1000 times. The shuffled dataset is used to calculate a shuffled probability of occurrence for each CIF and XPRA event. The original and shuffled probabilities are compared using a two-sided t-test with a standard significance level of 0.05. Since the goal of this test is to determine whether the projections are created through random chance, the two-sided t-test is used to compare the means between the projected and randomly generated dataset. Since this procedure is used for each cell in the data set, the probability of a null hypothesis being falsely rejected during the analysis of the results is high. To mitigate the

risk of overstating results, the procedure suggested by Wilks, 2016 is used to limit the false detection rate. This procedure creates a more sensitive null hypothesis threshold based on rank, number of tests, and a set control level for false detection rate. Assuming the results are spatially autocorrelated, a control level for false detection is set to two times the standard significance level (0.1) (Wilks, 2016).



$$\Pr(CIF|AR) = \frac{AR \cap CIF}{CIF} * 100\% \qquad \text{(Eq. 1)}$$

Each type of CIF event is characterized by calculating the frequency and magnitude of each detected event. Frequency of a CIF event is calculated as the total sum of events that occurred in each month. The magnitude of the CIF event refers to the total volume of surface runoff produced during the event itself or the day following it. Surface runoff is used as a measure of magnitude since it's a more direct indicator of flooding potential compared to rainfall alone. The average frequency and magnitude across all 28 members are used as a representation of the general trend associated with CIF events, while the

standard deviation across all 28 members is used as a representation of internal climate variability (ICV). The magnitude of CIF events without an AR present is also calculated (i.e., as non-AR event) to help contextualize the magnitude of the AR related CIF events.

## 3. Results

### 3.1 Evaluation of AR detection in CanRCM4-LE versus ERA5

To evaluate how well the CanRCM4-LE IVT outputs represent AR activity, the average number of ARs detected over the 31-year baseline period is calculated for each grid in both datasets. The CanRCM4-LE results are based on the 28-member ensemble mean. AR activity is expressed as the percentage of days within the 31-year period during which an AR occurs. AR activity detected in CanRCM4 compared to ERA5 is shown in Figure 2. Figure 2a considers the distribution of AR days in each region to give a better idea of how well CanRCM4 represents AR activity relative to ERA5. Results indicate that

CanRCM4 has a positive IVT bias in northern areas and a negative bias in southern areas. The positive bias is strongest in the PacificNW and Northern NA regions. The negative bias is strongest in the Desert region. The areas that experience a positive bias may overestimate AR related CIF events in future projections, while areas with a negative bias may underestimate them. Additionally, the central region displays higher positive biases in the dataset relative to the ERA5 and is a clear outlier. While the central region is outside the primary scope of this study, which focuses on coastal areas, this bias should be addressed in

future research. Despite these regional variations, the CanRCM4 distribution generally falls within the distribution detected in the ERA5 dataset. To examine how the general distribution of AR days changes over the whole study area, the average AR days was plotted with respect to latitude band inf Figure 2b. The overall difference between the CanRCM4-LE and observational reanalysis is low ranging from 0 to +/- 2.5%. These results also further highlight the apparent positive bias in results for more northern regions and negative bias in southern regions.



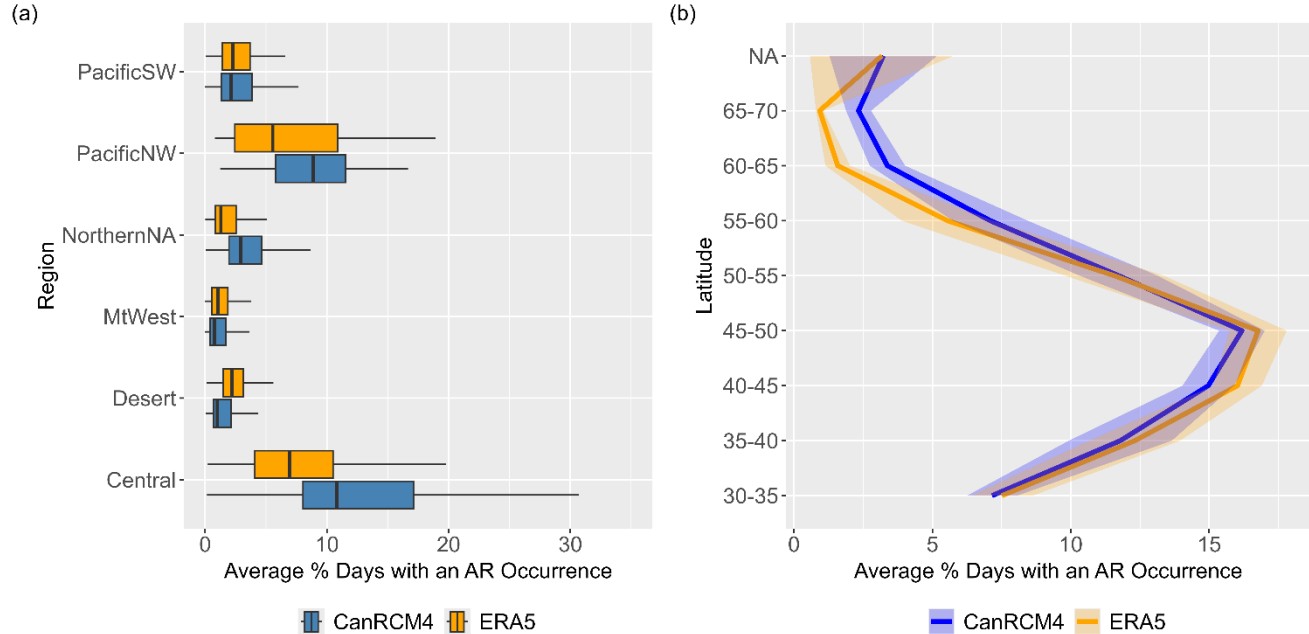


**Figure 2: Average AR occurrence in the Baseline Period(1986-2016) expressed as the overall percentage of AR days detected in the 31-year period for CanRCM4 and ERA5 models. The distribution of days in each region is shown in a), while the latitudinal average for each model over the entire study area is summarized in b) with the average spread in the results.**

## 3.2 Probability of Occurrence of a CIF being related to AR

To show the role ARs play in CIF events, the probability of an AR contributing to a CIF event is plotted spatially over the study area for each CIF event and warming level in Figure 3a Additionally, spatially averaged probabilities by latitudinal bands are presented in Figure 3b to provide a broader perspective. Figure 3 only displays results from the baseline period and WL4, the projections for all warming levels are included in Figure S1. These results show that ARs play a major role in most CIF events specifically in coastal and midlatitude regions. This is especially significant when considering the overall likelihood of CIF occurrence in relation to extreme runoff, as shown in Figure S2. The increases in likelihood of AR contribution to CIF events (Figure S1) mirror the increases in overall likelihood of CIF occurrence (Figure S2) highlighting the importance of ARs in CIF events.

The PacificNW region emerges as the most active area, with at least 90% or more CIF events attributed to AR activity in each cell. Projections indicate that ARs can have an increasing impact in a warmer climate, as seen in the gradual expansion of high probability areas across warming scenarios. As shown in the line plots at different latitudinal bands, AR contribution to CIF events is projected to steadily increase between warming periods. These projections are spatially consistent for SEF and XPRA events, however, spatial changes in Figure 3a suggest that for ROS events the biggest changes in likelihood occur in higher elevation areas. For example, the MtWest region, characterized by higher elevations as shown in Figure 1b, is projected to





experience an expansion in areas with a high likelihood of AR-related events. This aligns with findings from previous studies, which suggest that warmer climates may shift ROS events to higher altitudes (Il Jeong & Sushama, 2018; Musselman et al., 2018; Warden et al., 2024). It is important to note that in warmer climates, the statistical significance of projections decreases for ROS and SEF events, as shown by the reduction in stippling in the corresponding figures. This suggests that while ARs continue to play a significant role in extreme ROS and SEF events, these events may become less frequent or more challenging

to statistically detect under future warming scenarios. In contrast, XPRA events maintain higher statistical significance across warming levels, reflecting their more consistent occurrence. Despite this decrease in statistical significance, ARs remain critical drivers of ROS and SEF events, particularly in regions such as the PacificNW and MtWest. This is especially important considering Figure 1a which shows this projection affects a large portion of the dam infrastructure on the West Coast. The high likelihood in the PacificNW region in all event types stands out from the other regions. As noted in Section 3.1 this region

shows a positive bias for AR related activity. However, this high activity can also be attributed to this region containing most of the coastline in the study area and high elevation areas (see Figure 1b) which are both conducive to AR related storms. Therefore, although this area was found to have a positive bias relative to ERA, the high likelihood could also be caused regional characteristics that support AR related activity.

The latitudinal distribution of ROS and SEF events shown in Figure 3b peaks between 45°N and 55°N, with a secondary peak

at 60°N–65°N. The first peak corresponds to high-probability areas overlapping with the PacificNW and MtWest regions, where ARs penetrate furthest inland after landfall (Rutz et al., 2015). The secondary peak corresponds to an area of coastal Alaska not completely covered by the Bukovsky regions. In further regional breakdowns of the results this area will be referred to as the Western Alaska region. In contrast, XPRA events show a different latitudinal pattern, with a single peak occurring between 45°N and 50°N and declining further northward. In these peak regions ARs contribute heavily to ROS and SEF events,

and in some cases nearly all ROS and SEF events are AR-related. While XPRA events share some similarities, the overall probability of occurrence is lower, but the results exhibit higher statistical significance. This indicates that ARs may play less of a role in XPRA than CIF events. The second main difference is XPRA events show a different latitudinal pattern, with a single peak occurring between 45°N and 50°N and declining further northward. On the spatial maps this peak matches where ARs penetrate furthest inland as discussed previously.






**Figure 3: Spatial plot of probability of occurrence of an AR event given that ROS, SEF and XPRA events are occurring in the BL(left) and WL4(right) periods.. Statistically significant results are represented by areas with stippling. The line plots depict the spatial average of probability of occurrence for each latitudinal band and CIF event.**





### 3.3 Examining the Magnitude and Seasonality of AR related CIF Events


To examine the seasonality of AR related CIF events, the magnitude of these events is plotted in Figures 4-9. Figures 4-6 display the spatial average magnitude using bar plots in terms of rainfall, snowmelt and runoff in millimeters generated for each AR related events and events without AR influence (Regular Events). The spatial average is weighted to account for the distortion of different latitudes. The value displayed on the bar chart is a ratio calculated as the mean magnitude of AR related

events divided by the mean magnitude of regular events as a means of comparing the two scenarios. To further support these results, each cell with a non-zero magnitude is used to generate a kernel density plot for the study region in each season. This is to compare how likely a certain magnitude of an AR related event is relative to the likelihood of the magnitude of a regular event.

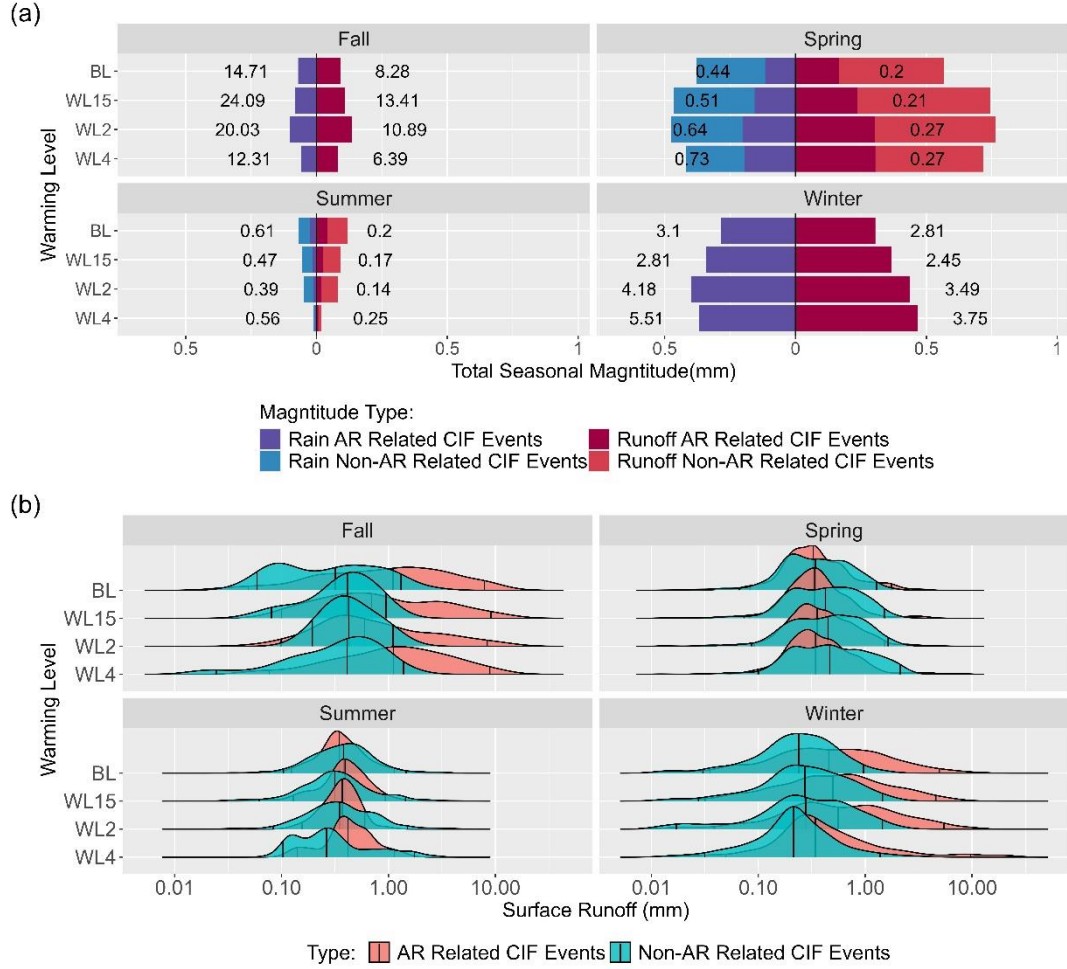

**Figure 4: Seasonal magnitude for ROS events. a) proportion of magnitude between AR and non AR events for each warming level in terms of precipitation (blue) and surface runoff (red). The values associated with each bar indicate the ratio of AR events to non AR related events as a more objective measure of the difference in magnitude. b) probability of magnitude expressed as surface runoff for each warming level for AR related events (red) and non-AR related events(green).**



Figure 4a shows that ARs are most influential in ROS events during the Fall and Winter months where they are shown to be
stronger by a large factor relative to non-AR events. However, unlike the probability of occurrence results, the magnitude is
not projected to increase consistently over time in any season except winter. Instead, there is a projected increase until WL2
and then a projected decrease. This projection is reflective of other studies (Il Jeong & Sushama, 2018; Musselman et al., 2018)
that also project increases in the frequency of ROS events in near term warming periods before decreasing in frequency in far
term warming periods. The rainfall amounts shown in blue reveal that changes in rainfall generally reflect any increases or
decreases in runoff magnitude. The kernel density plot for ROS events shown in Figure 4b shows that AR related ROS events
are more likely to have higher median and extreme magnitudes than regular ROS events especially in the Winter and Fall. AR
related ROS events are projected to be more likely stronger than regular events in future warming periods, as signified by the
upper and lower quantiles moving further to the right for all seasons, further supporting that ARs continue to contribute to the
highest magnitude ROS events.
Unlike ROS events, Figures 5 and Figure 6 show that SEF and XPRA events are projected to consistently increase in magnitude
with each warming scenario for all seasons. However, ARs contribute differently to each event. Figure 5a shows the ratio of
mean AR SEF magnitude to mean non-AR SEF magnitude is very high during seasons with peak AR activity. Whereas ROS
varied more seasonally, SEF magnitude is consistently projected to increase regardless of season, and by WL4, AR related
SEF events are projected to be stronger relative to regular events in most seasons except Spring. However, the difference
between AR related and regular events seems to decrease over time relative to the baseline period in terms of both generated
rainfall and runoff. This could be indicative of non-AR related events also becoming more extreme parallel to AR events.
However, the projections still show that AR related SEF events remain stronger than regular events in future warming
scenarios. The distribution graph in Figure 5b shows that AR related events are generally more likely to be higher in magnitude
relative to regular events in all seasons and warming periods. In a warmer climate AR related CIF events are more likely to
have higher median and extreme runoff magnitudes in all seasons indicated by a right ward shift in all distributions.




(a)

**Figure 5: Seasonal magnitude for SEF events. a) proportion of magnitude between AR and non AR events for each warming level in terms of precipitation (blue) and surface runoff (red). The values associated with each bar indicate the ratio of AR events to non AR related events as a more objective measure of the difference in magnitude. b) probability of magnitude expressed as surface runoff for each warming level for AR related events (red) and non-AR related events(green).**

The XPRA results are unique from the other two CIF events, since the projected runoff of AR related XPRA events is not as strong as the ROS and SEF events. This could be reflective of the results in section 3.2 which show that ARs contribute less to XPRA related flooding and supports the projected importance of precedent conditions in AR related flooding. This is shown





in Figure 6a, in which the displayed ratios for each category are very low relative to the CIF events discussed previously. Furthermore, the kernel density plots in Figure 6b indicate for most seasons, the runoff generated by AR related extreme precipitation can be expected to be lower. Despite these findings, Figure 7a also shows that the projected magnitude of XPRA events increase over time indicating that, like the CIF events, AR related extreme precipitation is also projected increase in severity in a warmer climate.

**Figure 6: Seasonal magnitude for XPRA events. a) proportion of magnitude between AR and non AR events for each warming level in terms of precipitation (blue) and surface runoff (red). The values associated with each bar indicate the ratio of AR events to non AR related events as a more objective measure of the difference in magnitude. b) probability of magnitude expressed as surface runoff for each warming level for AR related events (red) and non-AR related events(green).**

330                                                                 15



For a more in-depth analysis of the contributions of snowmelt, rainfall and runoff during AR related ROS events; the projected average rainfall and snowmelt on days of identified ROS events are plotted in a bar chart with the average runoff in Figure 7. This compares the approximate proportional contribution of snow and rainfall to ROS runoff. The change in the ratio of snowmelt to rainfall relative to the baseline period is plotted against event frequency in Figure 8 to further explore this

relationship. Figure 7 shows that, in general, changes in runoff across regions correspond proportionally to changes in the magnitude of rainfall and snowmelt associated with each event. Contrary to the expected decrease in snowmelt contribution in WL4, some regions may exhibit an increase in snowmelt contribution in the furthest projection. A notable example is the PacificNW region in winter, where the substantial increase in runoff likely skews the results in Figure 4a, leading to an apparent overall increase in winter event magnitude. Further examination of the PacificNW winter results suggests that this increased

average may be driven by a rise in event frequency rather than a shift in the relative contribution of snowmelt, which remains largely consistent with the baseline period. When comparing Figures 7 and 8, it becomes evident that regions experiencing an increase in runoff by WL4 also see an increase in event frequency, whereas regions with decreased runoff exhibit a corresponding decline in frequency. These findings are explored in depth relative to past studies in the following discussion of results.






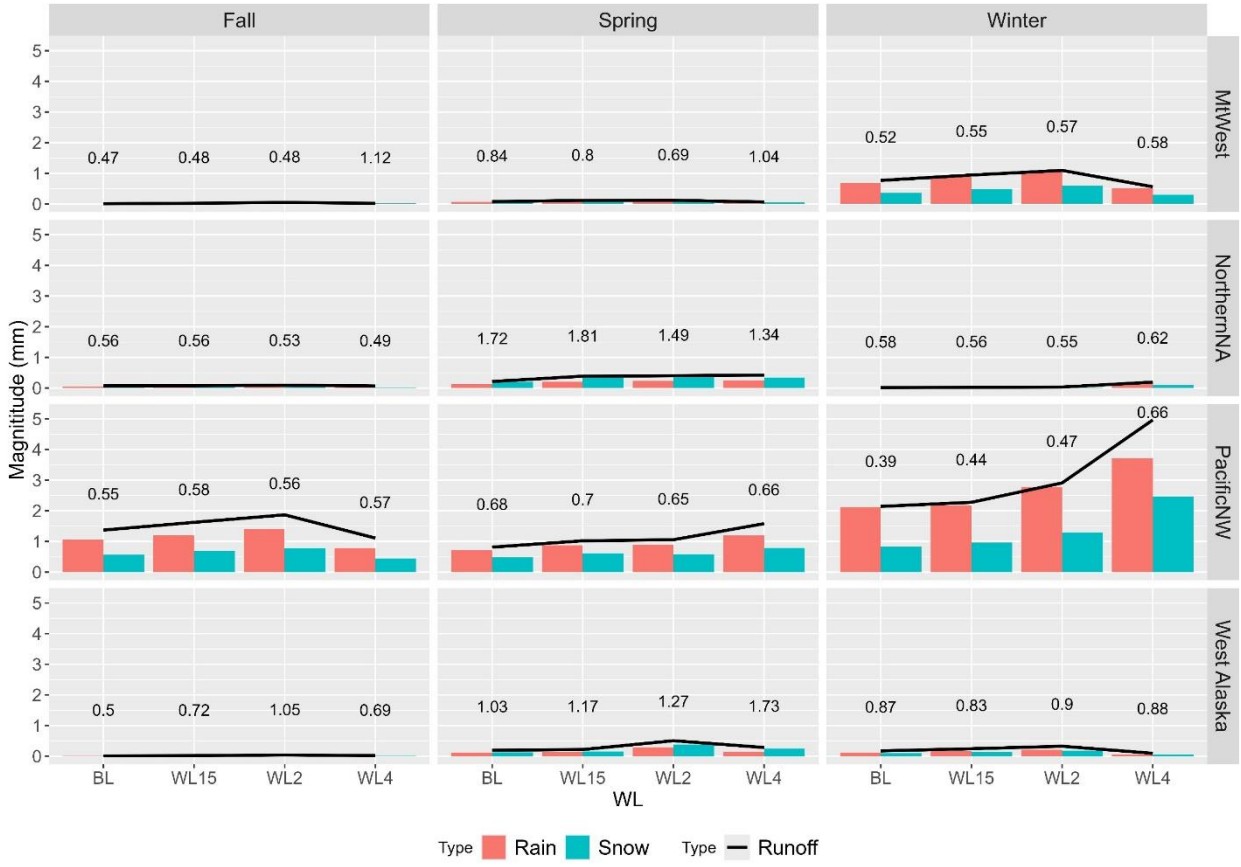

**Figure 7: The spatial average of AR related ROS event magnitude expressed in terms of millimetres of runoff, rainfall and snowmelt generated on the day of the event for each season and region. The left, middle and right columns depict results from fall, spring, and winter respectively. From the top, the first, second, third and fourth rows depict the MtWest, NorthernNA, PacificNW and West Alaska regions respectively.**




**Figure 8: The regional and seasonal difference in snowmelt to rainfall ratio for WL15 ,WL2, and WL4,relative to the baseline period versus ROS event frequency at each grid point. The left, middle and right columns depict results from fall, spring, and winter respectively. From the top, the first, second, third and fourth rows depict the MtWest, NorthernNA, PacificNW and West Alaska regions respectively. The black line in the center of each plot marks zero.**





To further examine regional seasonality, the month that contained the highest magnitude AR related event in terms of runoff is identified for each cell and plotted spatially in Figure 9. The findings in this figure support the results discussed above related to seasonality. For ROS events, the maps in the furthest warming period show that high impact events are projected to be more concentrated in winter seasons relative to the baseline period, especially in higher latitude areas. The results show a spatially homogenous projection which indicates ROS events will mainly occur in Winter in future warming scenarios. The seasonality of SEF and XPRA share a regional dependence that is unique relative to the ROS seasonality. Figure 9 clearly shows that northern regions experience AR related events in the fall while southern regions experience them in the winter. Furthermore, this seasonality remains consistent in the projected warming scenarios. Generally, these results show that during peak AR seasons ROS and SEF events on average have a higher magnitude than non-AR events. However, it is noted that changes in ROS events seem to be more closely related to factors such as elevation and snowpack levels. Furthermore, AR related CIF events are projected to be more likely and severe extremes. The difference in magnitude can largely be attributed to the increase in rainfall brought by a landfalling AR as indicated by the changes in precipitation results. This indicates that high runoff response is highly dependent on antecedent conditions.







**Figure 9: Spatial map showing the month with the highest ROS(top), SEF(middle) and XPRA(bottom) event magnitude in terms of runoff for each cell in the BL(left) and WL4(right) periods.**



## 3.4 Influence of Internal Climate Variability

The role of internal climate variability in the projection of AR related CIF events is examined using the Signal to Noise Ratio (SNR). The SNR is calculated by dividing the change in mean probability of occurrence relative to the base line period for each warming level by the standard deviation across all model members. Then SNR is calculated for each cell in the model and plotted spatially in Figure 10. A projected SNR that is greater than (less than) 1 (-1) indicates a significant positive (negative) change in the probability of an event being AR related relative to the base line period. The results suggest that significant signals (SNR > 1 or < -1) are not prominent until the farthest warming scenario (WL4). Even at WL4, the results vary substantially between event types, and the spatial patterns are noisier compared to the probability of occurrence maps shown in Section 3.2, especially in the lower coastal areas. This means that earlier findings that ARs will play a larger role in extreme CIF and precipitation events contains a significant amount of uncertainty. These results align with previous findings suggesting that projected AR activity in climate models is dominated by internal climate variability (Gershunov, et al., 2019; Tseng et al., 2022). This is also reflected in the probability of occurrence maps with a decrease in statistically significant results relative to randomly generated conditions in later warming periods. One of the recurring patterns between scenarios is the formation of stronger positive signals within the MtWest region and other higher region areas. For example, the ROS results project a strong positive signal emerging in the highest elevation areas that can be identified in Figure 1 in later warming levels. This shows areas where projections have the highest certainty that AR contributions to CIF and extreme precipitation events will increase. This projection is likely strong since AR related precipitation is often generated through orthographic uplift. These results show that internal variability creates significant uncertainty in the context of AR related compound flooding events.



**Figure 10: Signal to Noise Ratio for the probability that an event is AR related for each CIF event and extreme precipitation event. The SNR relative to the baseline period is shown for ROS, SEF and XPRA events in the top, middle and bottom rows respectively. Each column depicts a depicts a different warming level for each event, with the WL15, WL2, and WL4 being represented in the left, middle and right columns of plots respectively.**




## 4. Discussion

### 4.1 AR and CIF Event Interactions in a Changing Climate

Investigation of the likelihood of AR contribution to events revealed that ARs are major contributors to extreme ROS and SEF events, especially in coastal and orthographic areas. This is highlighted when considering the overall likelihood of CIF events

shown in Figure S2 which shows the areas with an increase in CIF likelihood generally see an increase in the likelihood of AR related events in Figure S1. Past studies in various places of North America also support that ARs disproportionately contribute to flooding events (Curry et al., 2019; Konrad & Dettinger, 2017; Nayak & Villarini, 2017). This relationship and its connection changing AR characteristics is explored in depth later in the discussion ARs contribute significantly less to overall extreme precipitation events compared to CIF events implying that AR related flooding is more dependent on pre-existing site

conditions and not the characteristics of a landfalling AR. This argument is supported by case studies of past AR flooding events showing that weaker more commonly occurring ARs could be associated with catastrophic flooding conditions (Gillett et al., 2022; Konrad & Dettinger, 2017; Ralph et al., 2006).   Another likely explanation for the results in this study is the increased amounts of ARs in the future will lead to an increase in temporally compounding atmospheric rivers. Temporally compounding ARs have been the subject of previous studies and would explain the high contribution of ARs to flooding events

that require precipitation related antecedent conditions (Bowers et al., 2023, 2024; Fish et al., 2022). In theory an AR could make landfall and create wet conditions or build up snowpack. A subsequent AR arriving shortly afterward could then generate significant runoff by interacting with the recently saturated soil or accumulated snowpack.

The seasonality of joint AR and CIF occurrence seems to be determined by the regional and seasonal behavior of ARs. This means that in southern regions along the West Coast such as Northern California, AR related CIF events are more likely to

occur in the winter. This is reflective of the seasonality commonly associated with ARs in the area (Gershunov et al., 2017; Ralph et al., 2013; Rutz et al., 2014). In higher latitudes like British Columbia or the Alaskan Coast, AR related events occur more in the fall or winter which is also consistent with past studies (Radić et al., 2015; Sharma & Déry, 2020a). This seasonality facilitates the occurrence of ROS, since ARs make landfall in colder season when there is more likely to be snowpack accumulated. Furthermore, in these regions the AR season occurs during the wet season which also promotes AR making

landfall in areas with high soil moisture. The observed seasonality further supports the finding that most CIF events are strongly connected to AR activity. AR related ROS events, in particular, align with the seasonality of AR activity but are more concentrated in the winter period, as ROS events require a certain amount of accumulated snowpack to occur. This seasonal concentration is also reflected in the magnitude results. The observed patterns could be influenced by expected changes in North America's climate, where winters are projected to become shorter and warmer, reducing overall snowpack accumulation

in many regions (Il Jeong & Sushama, 2018; IPCC, 2022; McCabe et al., 2007).

In a warmer climate, past studies showed that ARs are projected to become more frequent and severe (Espinoza et al., 2018; Gao et al., 2015; Radić et al., 2015; Shields & Kiehl, 2016). ARs are also projected to become warmer in temperature and are positively correlated with the warming of their area of origin (Gonzales et al., 2022). The increase in AR related CIF magnitude



and the likelihood of CIF events being driven by AR reflect these expected changes. Projections indicate that AR-related CIF
events will generally become more frequent in a warmer climate, particularly during winter. However, for an AR to produce
a high-magnitude CIF event, appropriate antecedent conditions, such as accumulated snowpack or saturated soils, are
necessary. This is highlighted through a comparison of magnitudes generated by AR related XPRA events to the other CIF
events. Although the CIF events are technically a subset of XPRA events, a strong connection with AR occurrence is only
seen once criteria for antecedent conditions are included. Due to the highly regional and seasonal behavior of ARs, the strong
connection between extreme surface runoff and ARs could help further strengthen flood frequency models. Especially since
numerous dams are in the areas that are projected to have the largest increases in AR related events. Furthermore, as a synoptic
scale phenomenon, ARs can be modelled more easily than other CIF related mesoscale phenomena (Lavers et al., 2016). Since
this relationship is also projected to strengthen in a warmer climate it could also be used as a basis for designing future
infrastructure against severe flooding events.

## 4.2 Identified Relationships between ARs and CIF Events

The ROS projections generally follow projections reported in existing literature ROS events are expected to become more
frequent and severe in higher latitudes and elevations in future warming scenarios (Beniston & Stoffel, 2016; Il Jeong &
Sushama, 2018; McCabe et al., 2007; Musselman et al., 2018). This is shown in the results through a projected increase in
average runoff between BL and WL2 until a decrease in WL4. This observation is often attributed to a warmer climate that
better facilitates the occurrence of rain during times when snowpack is present. Warmer ARs could further contribute to a
reduction in the amount of snowpack since the average temperature of AR related snowfalls are already close to the freezing
threshold, thus making them sensitive to changes in temperature (Gonzales et al., 2019, 2022; Guan et al., 2010). Although
increased AR frequency promotes the occurrence of ROS events in a warmer climate there may be less snow on the ground to
match this increase. One exception to this is during the winter in the PacificNW region where AR related ROS events are
projected to become more frequent and intense in the furthest warming period. This reflects the discussed changes in ARs,
which allow ROS events to occur in high elevation areas where snowpack is less affected by a warming climate. Past studies
have found these areas are often projected to experience an increase in ROS events due to higher likelihood of liquid
precipitation occurring during cold seasons in a warming climate (Beniston & Stoffel, 2016; McCabe et al., 2007; Warden et
al., 2024). As shown in Section 3.3 this increase in frequency is what drives an apparent increase in projected runoff. The
results of this study suggest that the occurrence of extreme ROS events could be facilitated by more frequent AR events in
future winters. This is further supported by the probability of occurrence maps, which indicate that a substantial portion of
ROS events in this region may be associated with AR activity. These findings imply that AR characteristics, such as frequency,
temperature, and intensity, are likely important factors influencing present and future extreme ROS events.

Saturated soil conditions are known for exacerbating flooding conditions (Grillakis et al., 2016; Wasko et al., 2020); however,
the results indicate that in a warmer climate a higher proportion of these events can be AR related. They further suggest that
this will be likely driven by the large amounts of precipitation delivered by a landfalling AR relative to events without AR





influence. Currently there is a lack of literature that explicitly examines how ARs contribute to high soil moisture levels. A recent study suggests that ARs reduce evapotranspiration due to an increased cloud cover associated with ARs (Chen et al., 2019). This could explain why the results suggest a strong connection between ARs and SEF events. However, whether there is a direct connection between AR conditions and SEF events generating runoff remains an open question. A more likely explanation is that ARs are coinciding with high soil moisture retained from other events. Furthermore, there is also the increased occurrence of temporally occurring ARs making landfall discussed previously. Together this evidence suggests that temporally compounding AR events and hydrologic autocorrelation could drive the occurrence of SEF flooding, which is further supported by the results obtained for individual heavy rainfall events. The results indicate that a single AR associated with heavy precipitation is less likely to generate large amounts of runoff compared to SEF and ROS events. Additionally, there appears to be a potential connection between the decrease in ROS events and an increase in SEF events. This relationship may be driven by higher winter temperatures, which reduce snowmelt-driven runoff (ROS events) while increasing rainfall, thereby contributing to more SEF events.

### 4.3 Analysis of Uncertainties

The projected AR related CIF events in this study are associated with considerable uncertainty, primarily due to internal variability. This level of uncertainty is not unexpected, as previous studies have shown that AR activity, extreme events, and precipitation are often more effected by internal variability (Blanusa et al., 2023; Gershunov et al., 2019; Kelleher et al., 2023; Mahmoudi et al., 2021). AR activity is dictated by thermodynamic changes, such as rising temperatures, and dynamic changes, like changes in jet stream position (Gao et al., 2015; Zavadoff & Kirtman, 2020). Changes in AR frequency and magnitude are dominated by thermodynamic changes, however, changes in landfall location are more attributable to changes in dynamic drivers (Gao et al., 2015; Tseng et al., 2022). Thermodynamic changes tend to exhibit stronger climate signals compared to dynamic changes, highlighting their dominant role in shaping AR characteristics under a warming climate (Deser et al., 2012). The established regional connection between AR occurrence and CIF events suggests that dynamic changes in AR behavior, such as shifts in landfall location, may play a dominant role compared to thermodynamic changes in shaping AR-related CIF events. If common landfalling locations shift significantly under future warming scenarios, this could explain the difficulty in establishing a statistically robust relationship between ARs and CIF events in later warming periods, as such shifts are not explicitly accounted for in the current methodology. This highlights the need for further research to reduce uncertainties in modeling AR dynamics, which could improve projections of AR-related CIF events and their associated impacts under future climate scenarios (Gershunov, et al., 2019; Tseng et al., 2022). Increasing the number of members used in a model ensemble has been shown to reduce the uncertainty related to internal variability in climate change studies (Deser, 2020; Deser et al., 2012) especially in the context of compound events (Bevacqua et al., 2023). However, similar studies examining internal variability and ARs have had similar results despite having a higher number of members(Huang & Swain, 2022; Tseng et al., 2022) and compound inland flooding events have also been found to be strongly associated with internal variability(Fereshtehpour et al., 2025). It is also important to note that significant uncertainty exists in the characterization of



ARs themselves, largely due to the lack of consensus on how ARs should be defined as objects (Collow et al., 2022; Lora et al., 2020; Zhou et al., 2021).

**5. Conclusions**

ARs have often been associated with catastrophic flooding events; however, past studies often focused on these events though a univariate lens by focusing on extreme precipitation or streamflow. Studies that do focus on AR interactions with preexisting

conditions often overlook the potential impacts of climate change. The goal of this study is to quantify the extent to which CIF events can be attributed to ARs on the North American West Coast and assess how this relationship is projected to change under future climate scenarios. Using outputs from the CanRCM4-LE model, the analysis focused on calculating the projected contributions of ARs to CIF events, as well as the projected frequency and magnitudes of AR-driven CIF events. The results indicate that ARs are expected to play a growing role in the occurrence and severity of extreme ROS and SEF events. In

contrast, ARs are found to have a less pronounced role in heavy rainfall events leading to the heavy runoff, highlighting the importance of antecedent conditions in AR related flooding events. This underscores the value of incorporating landfalling ARs within a compound event framework. In a warmer climate a majority of extreme ROS and SEF events may be associated with landfalling ARs. The seasonality of landfalling ARs, as identified in previous studies, complements and shapes the seasonality of AR-driven CIF events. The results regarding projected flooding risk and changes in CIF events align closely

with past works and case studies. These findings have implications for the prediction and analysis of future extreme runoff events and the design of critical infrastructure. Furthermore, they highlight that considering the impact of an AR by itself can underestimate its impact, and the role of in-situ conditions must be considered along with AR strength when considering potential impacts. Despite these insights, the results carry considerable uncertainty, primarily due to internal climate variability, the exclusion of dynamic factors, sample size limitations, and AR detection methods. Future studies can improve the

methodology by focusing on more characteristics of ARs such as specific AR intensity category and the role of sequential ARs. Another potential area of future focus is the role ARs play in generating extreme precipitation in comparison to convective mechanisms. Reducing uncertainty in future analyses may involve accounting for potential dynamic changes in AR positioning and employing multiple AR detection methods to better quantify how much uncertainty is associated with the definition of an AR.

**Author Contribution**

AG: Conceptualization, Data curation, Formal analysis, Investigation, Methodology, Software, Validation, Visualization, Writing – original draft; MF: Conceptualization, Data curation, Formal analysis, Investigation, Methodology, Writing – review & editing; MN: Conceptualization, Formal analysis, Investigation, Methodology, Software, Resources, Validation, Writing –



review & editing, Funding acquisition, Project administration; AC: Data curation, Investigation, Writing – review & editing;
HS: Investigation, Resources, Writing – review & editing..

### Acknowledgments

This research was supported by Environment and Climate Change Canada (ECCC) through the Flood Hazard Identification and Mapping Program (Grant GCXE25M006).

### Data Availability Statement

The data from the Canadian Regional Climate Model Large Ensemble (CanRCM4 LE) used in this study can be accessed on the Government of Canada Open Data Portal at: https://crd-data-donnees-rdc.ec.gc.ca/CCCMA/products/CanSISE/output/CCCma/CanRCM4/. The ERA5 reanalysis dataset used for validation in this study is available at the ECWMF climate data store which can be accessed at: https://cds.climate.copernicus.eu/datasets. The elevation data from the Global Multi-resolution Terrain Elevation Data 2010 can be accessed from the United States Geological
Survey earth explorer data portal at https://earthexplorer.usgs.gov/. The Global Dam tracker data base can be found in cited paper within the manuscript or publicly available at https://zenodo.org/records/7616852.

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
