# Peer review of "Atmospheric Rivers as Triggers of Compound Flooding: Quantifying Extreme Joint Events in Western North America Under Climate Change"

_EGUsphere, 2025_

## Author Comment (AC1)

**Editor**

We would like to thank the editor and reviewers for their constructive feedback. Their comments have greatly helped improve the clarity and overall quality of the manuscript. In this revised version, we have strengthened the methodological explanations, clarified definitions and event-detection criteria, expanded the discussion of uncertainties and sensitivity analyses, and refined the figures and references to enhance readability and consistency with journal guidelines.

**Reviewer #1**

**General Comment:**

This study examines how atmospheric rivers (ARs) contribute to inland flooding by looking at how often ARs, extreme rainfall, and compound events like Rain on Snow (ROS) and Saturation Excess Flooding (SEF) occur together along the western coast of North America. The authors also study how these events change with seasons and how future warming from climate change and natural variability could affect them. While the topic is important and the study includes some new findings, there are serious concerns about the methods used, especially how ARs and compound events are identified and defined. The authors need to address the following points before the manuscript is ready for submission to a scientific journal.

We thank the reviewer for their constructive feedback and recognition of the study's relevance. In this revision, we have clarified the event detection methods, strengthened methodological justification, and added supporting sensitivity analyses to address the reviewer's concerns.

**Major comments:**

- 1. The authors detect ARs using a fixed IVT threshold of 250 kg/m/s that persists for more than 24 hours at a single grid cell. However, this approach does not consider the typical structure of ARs which are long narrow bands of water vapor stretching thousands of kilometers. Many studies including Guan and Waliser 2015; 2019 have shown that ARs should be identified based on both moisture intensity and their spatial shape such as length and width. Ignoring these features can lead to the inclusion of unrelated weather systems that temporarily meet the IVT threshold but do not have the structure of an AR.
- 2. Using only a point-based threshold without checking for spatial coherence may result in misclassification of short or wide moisture plumes (isolated moisture plumes, mesoscale convective systems, or tropical plumes) as ARs. This is especially concerning because ARs are known for their coherent filament-like structure and influence over large regions. Relying only on how long the IVT exceeds a threshold at a single location does not ensure that a true AR is present. It is unclear how the authors distinguish between real ARs and other systems that happen to bring strong moisture transport for a day or more.

**Response #1&2:**

We appreciate the reviewer's thoughtful comments. Regarding AR structure, we acknowledge that ARs are coherent elongated structures. However, the primary objective of this study is to examine compound inland flooding events in midlatitude regions of North America, rather than to characterize AR morphology in detail. Accordingly, a point-based IVT threshold approach was adopted as a fit-for-purpose and methodologically defensible choice. This approach is also computationally efficient for large-ensemble simulations, where applying full spatial coherence criteria (e.g., length and width filters) would substantially increase computational cost and limit the feasibility of our event-based analyses.

We agree that ARs are typically characterized by both moisture intensity and spatial coherence and that several widely used algorithms incorporate length and width constraints to better represent these features. To address this concern, we refer to the sensitivity analysis conducted by Guan and Waliser (2023), who compared their improved 2019 algorithm with a point-based IVT threshold method (similar to that employed in our study). They noted that:

"the overall pattern similarity between results based on the simple algorithm methods outside the tropics supports the simplicity of the AR definition adopted in the original AR scale, which can be especially helpful for examination of ARs using point-based data (e.g., from in situ observations) or gridded data over a small domain (e.g., from regional model simulations), for which geometrical considerations of ARs cannot be readily applied." (Guan et al., 2023)

Our approach is thus consistent with established studies using regional climate models (e.g., Chen et al., 2019; Radić et al., 2015; Sharma and Déry, 2020). It strikes a balance between physical realism and computational efficiency, providing a robust means of identifying periods of intense moisture transport relevant to inland flooding. Additional comparisons with other algorithms are provided in Figure R2 and discussed in Response #4, confirming the consistency of results across detection methods.

3. Although the study references Guan and Waliser 2015;2019, it does not apply their well-known AR detection method which includes geometry and landfall criteria. It is important to ask why the authors cite this work without adopting its key methods. Moreover, the fixed IVT threshold may not work equally well across different regions or seasons. In fact, Ralph et al 2019 and others have suggested that thresholds should be adjusted based on local climate and conditions. There is also no discussion of whether the chosen threshold or duration was tested for sensitivity.

**Response #3:**

We thank the reviewer for this insightful observation. The works of Guan and Waliser (2015; 2019) are cited because they represent foundational contributions to AR research. However, our study focuses on the co-occurrence of ARs and compound inland flooding rather than detailed AR morphology. For this reason, we employed a point-based IVT threshold approach, which has been widely applied in studies across western North

America (e.g., Chen et al., 2019; Radić et al., 2015; Rutz et al., 2014; Sharma and Déry, 2020a, b). Additionally, we have included a sensitivity analysis below in Figure R1. This analysis evaluates the effect of varying IVT thresholds on the identification of moisture transport events. The spatial patterns of AR-related ROS occurrence remain consistent across thresholds, confirming the robustness of our findings to reasonable parameter variations. While localized calibration may further refine detection for specific regions or seasons, the adopted threshold captures the large-scale patterns relevant to inland flooding in this study. Future work will extend this framework to include geometry-based AR detection and seasonal threshold optimization.

Figure R1: Sensitivity analysis for different IVT thresholds. The plots show the probability that an ROS event co-occurs with an AR event for an IVT threshold of 500kg/m/s(left), 250kg/m/s(middle), and 100kg/m/s(right) given that an ROS event is occurring.

4. In recent years several objective and widely accepted methods have been developed to detect ARs. These methods have been compared through efforts like the ARTMIP project. The study would benefit from comparing their results with at least one such established method.

**Response #4:**

We thank the reviewer for this valuable suggestion. We agree that multiple AR detection algorithms exist, and recent intercomparison projects such as ARTMIP have highlighted the diversity of approaches and their influence on AR climatologies. The primary objective of this study, however, is to assess the linkage between ARs and compound inland flooding under climate change rather than to optimize the performance of specific detection algorithms.

To assess methodological robustness, we conducted a targeted sensitivity test comparing two widely used algorithms, TEMPEST and Guan & Waliser (2019), with the point-based IVT threshold method employed in this study (Figure R2). The results demonstrate broadly consistent regional patterns of AR frequency across methods, supporting the suitability of the chosen approach for this application. Future work will further explore algorithmic differences and identify the most appropriate detection frameworks for compound inland flood analyses under varying climate conditions.

Figure R2: Average AR occurrence in the Baseline Period (1986-2016) expressed as the overall percentage of AR days detected in the 31-year period for ERA5. The distribution of days detected in each region in the current study is compared for the Guan and Waliser algorithm, the point-based threshold algorithm and the TEMPEST algorithm. The Guan and Waliser algorithm results represent ARs counted on a daily and sub daily scale.

5. The authors mention that they selected the Eulerian method for compatibility with CIF detection. However, this seems more like a convenience than a scientific justification. ARs are large scale features, and it is important to detect them based on their full structure rather than just point based conditions. This would likely provide a more realistic match to compound flood events which also involve broad spatial atmospheric processes.

**Response #5:**

We thank the reviewer for this comment. Our analysis focuses on the temporal cooccurrence of ARs and compound inland flooding events, rather than on the detailed
spatial geometry of ARs. The Eulerian, point-based approach was therefore selected to
ensure consistency with the CIF detection framework, which is also defined at fixed grid
locations and aggregated across large ensembles. This alignment allows us to
systematically link periods of intense moisture transport to flood events across the study
domain. While object-based methods that incorporate AR length, width, and landfall
characteristics are well suited for studies of AR morphology and evolution, their added
value here would be limited given our focus on temporal coincidence rather than spatial
extent. Future studies may complement this work by applying geometry-based AR
detection to assess spatial clustering of compound events and localized hydrological
responses.

6. In the introduction, SEF events are described as resulting from pre-existing soil saturation before intense rainfall. However, in the methodology, SEF events are defined based on co-occurrence of soil moisture and precipitation exceeding the 98th percentile on the same day. This contradicts the stated physical mechanism of SEF. It would be more appropriate to consider soil moisture on the day prior to precipitation, which better reflects antecedent saturation conditions driving saturation-excess runoff.

**Response #6:**

As noted, the antecedent soil moisture is the key driver of saturation-excess flooding. In our modeling framework, the soil moisture variable used represents conditions that are effectively inherited from the preceding day(s), since it is updated based on cumulative infiltration and evapotranspiration. Therefore, high soil moisture values observed on the day of heavy precipitation can be interpreted as reflecting antecedent saturation conditions. With both soil moisture and precipitation to exceed the 98th percentile on the same day, our approach ensures that SEF events are identified when (a) the catchment is already near saturation and (b) an intense rainfall event occurs while this saturated state persists. Using the previous day's soil moisture could also be a valid approach and more conservative, but given the daily resolution of our data, this would not materially change the interpretation of antecedent saturation. We will clarify this methodological point in the revised manuscript to avoid possible confusion for readers.

7. The methodology does not clarify whether consecutive extreme precipitation days are treated as a single event or as multiple separate events. Multiday rainfall can lead to sustained soil saturation and progressive runoff generation. Defining such events based on their onset date and grouping them accordingly would align better with the hydrological reality and reduce double-counting.

**Response #7:**

In this study, extreme precipitation days are defined as when the 98th percentile threshold of rain is exceeded resulting in runoff that exceeds the 98th percentile threshold on or one day after the event. With this methodology consecutive precipitation days are treated as a single event.

We note that this work is a companion to Fereshtehpour et al. (2025), where the sensitivity of results to grouping of consecutive days and onset-date definitions was explicitly tested. To further justify this choice, we refer to Fereshtehpour et al. (2025), where a sensitivity analysis was performed using ERA5 data to explore various gap-day criteria for identifying independent compound events. Specifically, consecutive events were merged if separated by up to 0, 1, 2, or 3 days, with the zero-gap case representing the baseline approach used here. The analysis (Supplementary Text S2 and Figure S7 in Fereshtehpour et al., 2025) showed that the spatial patterns and overall statistics of events remained largely consistent across all gap settings, indicating that treating consecutive

days as separate events does not materially affect the identification of compound flooding events. Therefore, we consider the current definition to be appropriate for the large-scale, statistical nature of this analysis. We will clarify this point in the revised manuscript.

8. Similarly, the runoff condition (98th percentile on the same day or the next) should be evaluated over the entire event window, not just individual days, particularly in cases of compound events where runoff can build up over multiple days.

**Response #8:**

This approach is consistent with the daily event definition adopted in our study. We accounted for the concern raised by the reviewer by extending the hydrological response timeframe to include both the day of the precipitation/extreme soil moisture event (day d) and the following day (d + I). This means that the associated hydrological responses to a precipitation event typically unfold over approximately two days. This extension reflects the fact that, when working with daily data, the majority of runoff generation and soil moisture response to intense precipitation typically occurs within this two-day window. As a result, our methodology captures both the immediate and slightly delayed hydrological responses while remaining compatible with the temporal resolution of the dataset.

9. ROS events are defined as days when both daily precipitation and snowmelt exceed the 98th percentile, with snowmelt contributing at least 20% of total liquid water. However, the approach raises a key question: can sub-98th percentile multiday rainfall still produces substantial snowmelt that satisfies the 20% contribution threshold? If so, limiting detection to only those days with extreme precipitation may exclude physically valid ROS events.

**Response #9:**

Thank you for this comment. Our use of the 98th-percentile precipitation threshold was a deliberate choice to focus on high-impact CIF events, consistent with the primary objective and scale of this study. This approach may exclude some physically valid but less extreme ROS events, particularly those involving moderate but persistent rainfall on snow. This exclusion is by design, as our goal was to characterize the most consequential events rather than to capture the full climatology of ROS occurrences. In the revised manuscript, we will explicitly discuss this methodological choice and its implications.

10. The ROS definition also appears sensitive to daily co-occurrence of rain and snowmelt. Yet in reality, snowmelt lag and meltwater routing processes might cause peak melt to occur slightly after rainfall, particularly in colder regions. Has this temporal mismatch been accounted for or tested?

**Response #10:**

A one-day runoff extension is included in the definition of detected compound flooding events to capture short lags between rainfall and meltwater response within daily frameworks. We did not

explicitly test different lag times in the present study, as the objective was to evaluate large-scale trends in compound events and their potential connection with atmospheric rivers, rather than to reproduce event-level hydrological timing in detail. An approach that considers longer lag times would require a process-based modelling setup that can consider snowmelt dynamics. However, this methodology is beyond the scope of the large-scale statistical analysis of the current work. We will clarify this methodological assumption in the revised manuscript and note this as an important direction for future work.

11. The use of fixed 98th percentile thresholds across all variables, grid cells, and time frames may not equally capture extremes in diverse hydroclimatic settings. Have the authors performed any regional or seasonal sensitivity tests to ensure that this uniform threshold is not excluding impactful events in drier or colder basins?

**Response #11:**

The 98th percentile thresholds in our study are computed individually for each grid cell based on the local climatology, rather than applied as a single uniform value across the domain. This approach ensures that the definition of "extreme" is region-specific and reflects the hydroclimatic variability across wet, dry, warm, and cold basins. In addition, threshold sensitivity was examined in *Fereshtehpour et al.* (2025) for a similar analysis, where varying the percentile (e.g., 95th–99th) produced consistent spatial patterns and trends. Please see figure below. This supports the robustness of the 98th percentile choice for capturing impactful events across diverse regions. We will clarify this point in the revised manuscript to avoid potential misunderstanding.

Figure 3: Probability of occurrence (%) of ROS events under different thresholds (0.9-0.99) for rainfall and snowmelt (Fereshtehpour et al., 2025; Figure 6 in SI).

12. It is unclear whether non-snow season months were filtered out before detecting ROS events. Including periods when snowmelt is physically implausible might generate false positives or inflate event counts due to random high precipitation days.

**Response #12:**

Non-snow season months are implicitly excluded by our definition: the 20% snowmelt contribution threshold removes days with zero or negligible snowmelt, so that high-precipitation days without snowmelt do not generate false ROS detections. Moreover, events are first detected on a daily basis and then aggregated by month and season, so the seasonal statistics reflect only periods when both precipitation and snowmelt jointly contribute. We will clarify this point in the revised manuscript.

13. The method used to associate ARs with CIF events is based on day-of co-occurrence. Given the possible lag between AR landfall and inland flooding, particularly in snow-dominated regions, it would be helpful to test a wider time window (e.g.,  $\pm 1$  or 2 days) when evaluating AR-related CIFs.

**Response #13:**

In our framework, CIF events are defined based on daily precipitation and hydrological responses; therefore, aligning AR detection to the same day ensures consistency with the event definition. We should highlight that CIF events are identified based on cooccurring precipitation and hydrological state (soil moisture, snowmelt), not solely on runoff magnitude. While it is true that inland flooding peaks can lag AR landfall, particularly in snow-affected basins, this lag is already partly accounted for through the inclusion of runoff response on day d and d+1. Associating ARs to the same day as precipitation avoids disconnecting the meteorological driver (AR) from the hydrological trigger (precipitation) that defines the event.

Additionally, AR-related extreme precipitation is most commonly associated within a ±24-hour (same-day) window, which has been shown to capture the majority of hydrologically significant AR impacts (Chen et al., 2019; Oakley et al., 2017; Sharma and Déry, 2020b). This practice is further supported by case studies of impactful events (Gillett et al., 2022; Michaelis et al., 2022; Ralph et al., 2006).

**Minor Comments:**

1. The authors may consider briefly mentioning whether there are other notable moisture transport mechanisms in Western North America besides atmospheric rivers. If so, a short explanation in the introduction would provide useful context.

**Response #1:**

Atmospheric rivers are indeed recognized as the dominant moisture transport mechanism delivering extreme precipitation to much of western North America. ARs have been found to be dominate contributors to the frequency and intensity of flooding events (Dettinger, 2016; Lamjiri et al., 2018; Rutz et al., 2014). Other large-scale mechanisms, such as cut-off lows (Mishra et al., 2025) and synoptic-scale frontal systems (Pryor et al., 1995), can also transport moisture inland, but their contributions are generally less frequent or less intense compared to ARs. ARs are also typically embedded within synoptic scale frontal systems connected with extratropical cyclones (Ralph et al., 2020). Our focus on ARs is motivated by their well-documented role as the primary source of moisture for high-impact winter storms and their established connection to flooding hazards in this region.

In the revised manuscript, we will explicitly acknowledge these additional moisture transport mechanisms in the introduction to provide context while emphasizing that ARs are the primary focus of this work due to their dominant role in driving high-impact precipitation and flood events across this region.

2. In the introduction, it is unclear what processes contribute to pre-existing soil saturation prior to Saturation Excess Flooding (SEF) events. Is the saturation driven by earlier low-intensity ARs, local convective rainfall, or antecedent snowmelt before the landfall of stronger ARs? Clarification on the dominant mechanisms would strengthen the physical interpretation.

**Response #2:**

Section 4.2 discusses the role of temporally compounding ARs as a potential key preconditioning factor. Antecedent soil condition also often arises from moderate rainfall or snowmelt as well, both of which have also been shown to be associated with ARs. However, our work does not definitively identify a dominant mechanism.

Given the spatial heterogeneity of moisture sources and the complexity of soil moisture dynamics, a comprehensive attribution of pre-saturation mechanisms would require a dedicated study combining AR detection, storm tracking, and hydrological modeling. We will clarify in the revised introduction that multiple processes can contribute to antecedent saturation, while highlighting the potential importance of AR sequences in preconditioning soils for SEF events. In future studies process level hydrologic modelling should be used to properly identify the main driver of SEF events.

3. While a citation is provided for the bias-corrected dataset, the manuscript does not specify which climate variables were bias corrected. Please list the variables for clarity.

**Response #3:**

We appreciate the reviewer's attention to this point. To clarify, the climate variables in this study were from the non bias corrected CanRCM4-LE simulations. We will revise the manuscript to state explicitly that no bias-corrected variables were used in our analysis.

**References**

Bowers, C., Serafin, K. A., and Baker, J. W.: Temporal compounding increases economic impacts of atmospheric rivers in California, Sci. Adv., 10, eadi7905, https://doi.org/10.1126/sciadv.adi7905, 2024.

Chen, X., Leung, L. R., Wigmosta, M., and Richmond, M.: Impact of Atmospheric Rivers on Surface Hydrological Processes in Western U.S. Watersheds, J. Geophys. Res. Atmospheres, 124, 8896–8916, https://doi.org/10.1029/2019JD030468, 2019.

Dettinger, M.: Historical and Future Relations Between Large Storms and Droughts in California, San Franc. Estuary Watershed Sci., 14, https://doi.org/10.15447/sfews.2016v14iss2art1, 2016.

Fereshtehpour, M., Najafi, M. R., and Cannon, A. J.: Characterizing Compound Inland Flooding Mechanisms and Risks in North America Under Climate Change, Earths Future, 13, e2024EF005353, https://doi.org/10.1029/2024EF005353, 2025.

- Gillett, N. P., Cannon, A. J., Malinina, E., Schnorbus, M., Anslow, F., Sun, Q., Kirchmeier-Young, M., Zwiers, F., Seiler, C., Zhang, X., Flato, G., Wan, H., Li, G., and Castellan, A.: Human influence on the 2021 British Columbia floods, Weather Clim. Extrem., 36, 100441, https://doi.org/10.1016/j.wace.2022.100441, 2022.
- Guan, B., Waliser, D. E., and Ralph, F. M.: Global Application of the Atmospheric River Scale, J. Geophys. Res. Atmospheres, 128, e2022JD037180, https://doi.org/10.1029/2022JD037180, 2023.
- Hagos, S. M., Leung, L. R., Yoon, J.-H., Lu, J., and Gao, Y.: A projection of changes in landfalling atmospheric river frequency and extreme precipitation over western North America from the Large Ensemble CESM simulations, Geophys. Res. Lett., 43, 1357–1363, https://doi.org/10.1002/2015GL067392, 2016.
- Lamjiri, M. A., Dettinger, M. D., Ralph, F. M., Oakley, N. S., and Rutz, J. J.: Hourly analyses of the large storms and atmospheric rivers that provide most of California's precipitation in only 10 to 100 hours per year, San Franc. Estuary Watershed Sci., 16, 1–17, https://doi.org/10.15447/sfews.2018v16iss4art1, 2018.
- Michaelis, A. C., Gershunov, A., Weyant, A., Fish, M. A., Shulgina, T., and Ralph, F. M.: Atmospheric River Precipitation Enhanced by Climate Change: A Case Study of the Storm That Contributed to California's Oroville Dam Crisis, Earths Future, 10, e2021EF002537, https://doi.org/10.1029/2021EF002537, 2022.
- Mishra, A. N., Maraun, D., Schiemann, R., Hodges, K., Zappa, G., and Ossó, A.: Long-lasting intense cut-off lows to become more frequent in the Northern Hemisphere, Commun. Earth Environ., 6, 115, https://doi.org/10.1038/s43247-025-02078-7, 2025.
- Oakley, N. S., Lancaster, J. T., Kaplan, M. L., and Ralph, F. M.: Synoptic conditions associated with cool season post-fire debris flows in the Transverse Ranges of southern California, Nat. Hazards, 88, 327–354, https://doi.org/10.1007/s11069-017-2867-6, 2017.
- Pryor, S. C., McKendry, I. G., and Steyn, D. G.: Synoptic-Scale Meteorological Variability and Surface Ozone Concentrations in Vancouver, British Columbia, 1995.
- Radić, V., Cannon, A. J., Menounos, B., and Gi, N.: Future changes in autumn atmospheric river events in British Columbia, Canada, as projected by CMIP5 global climate models, J. Geophys. Res. Atmospheres, 120, 9279–9302, https://doi.org/10.1002/2015JD023279, 2015.
- Ralph, F. M., Neiman, P. J., Wick, G. A., Gutman, S. I., Dettinger, M. D., Cayan, D. R., and White, A. B.: Flooding on California's Russian River: Role of atmospheric rivers, Geophys. Res. Lett., 33, https://doi.org/10.1029/2006GL026689, 2006.
- Ralph, F. M., Wilson, A. M., Shulgina, T., Kawzenuk, B., Sellars, S., Rutz, J. J., Lamjiri, M. A., Barnes, E. A., Gershunov, A., Guan, B., Nardi, K. M., Osborne, T., and Wick, G. A.: ARTMIPearly start comparison of atmospheric river detection tools: how many atmospheric rivers hit northern California's Russian River watershed?, Clim. Dyn., 52, 4973–4994, https://doi.org/10.1007/s00382-018-4427-5, 2019.

- Ralph, F. M., Dettinger, M. D., Jonathan, R. J., and Waliser, D. E. (Eds.): Atmospheric Rivers, 1st ed., Springer International Publishing, 284 pp., 2020.
- Ridder, N., De Vries, H., and Drijfhout, S.: The role of atmospheric rivers in compound events consisting of heavy precipitation and high storm surges along the Dutch coast, Nat. Hazards Earth Syst. Sci., 18, 3311–3326, https://doi.org/10.5194/nhess-18-3311-2018, 2018.
- Rutz, J. J., Steenburgh, W. J., and Ralph, F. M.: Climatological Characteristics of Atmospheric Rivers and Their Inland Penetration over the Western United States, Mon. Weather Rev., 142, 905–921, https://doi.org/10.1175/MWR-D-13-00168.1, 2014.
- Rutz, J. J., Shields, C. A., Lora, J. M., Payne, A. E., Guan, B., Ullrich, P., O'Brien, T., Leung, L. R., Ralph, F. M., Wehner, M., Brands, S., Collow, A., Goldenson, N., Gorodetskaya, I., Griffith, H., Kashinath, K., Kawzenuk, B., Krishnan, H., Kurlin, V., Lavers, D., Magnusdottir, G., Mahoney, K., McClenny, E., Muszynski, G., Nguyen, P. D., Prabhat, Mr., Qian, Y., Ramos, A. M., Sarangi, C., Sellars, S., Shulgina, T., Tome, R., Waliser, D., Walton, D., Wick, G., Wilson, A. M., and Viale, M.: The Atmospheric River Tracking Method Intercomparison Project (ARTMIP): Quantifying Uncertainties in Atmospheric River Climatology, J. Geophys. Res. Atmospheres, 124, 13777–13802, https://doi.org/10.1029/2019JD030936, 2019.
- Sharma, A. R. and Déry, S. J.: Contribution of Atmospheric Rivers to Annual, Seasonal, and Extreme Precipitation Across British Columbia and Southeastern Alaska, J. Geophys. Res. Atmospheres, 125, e2019JD031823, https://doi.org/10.1029/2019JD031823, 2020a.
- Sharma, A. R. and Déry, S. J.: Linking Atmospheric Rivers to Annual and Extreme River Runoff in British Columbia and Southeastern Alaska, J. Hydrometeorol., 21, 2457–2472, https://doi.org/10.1175/JHM-D-19-0281.1, 2020b.
- Tseng, K.-C., Johnson, N. C., Kapnick, S. B., Cooke, W., Delworth, T. L., Jia, L., Lu, F., McHugh, C., Murakami, H., Rosati, A. J., Wittenberg, A. T., Yang, X., Zeng, F., and Zhang, L.: When Will Humanity Notice Its Influence on Atmospheric Rivers?, J. Geophys. Res. Atmospheres, 127, e2021JD036044, https://doi.org/10.1029/2021JD036044, 2022.

---

## Author Comment (AC2)

**Reviewer #2:**

This is a welcome contribution to the literature on atmospheric rivers as triggers of compound flooding. However, it is unclear if sufficient work has been undertaken at this stage to justify publication now.

The authors note that: "If common landfalling locations shift significantly under future warming scenarios, this could explain the difficulty in establishing a statistically robust relationship between ARs and CIF events in later warming periods, as such shifts are not explicitly accounted for in the current methodology. This highlights the need for further research to reduce uncertainties in modeling AR dynamics."

The authors are to be commended for admitting in the conclusions that: "The results carry considerable uncertainty, primarily due to internal climate variability, the exclusion of dynamic factors, sample size limitations, and AR detection methods. Future studies can improve the methodology by focusing on more characteristics of ARs."

For those involved in flood risk decision-making, the paper in its present form is much less informative and useful than it might be if further research would be undertaken to address some of the key uncertainties identified by the authors themselves.

**Response:**

We sincerely thank the reviewer for the constructive feedback. We organized our response around the main criticisms raised as follows:

- (a) Justification of the study's contributions
- (b) Clarification of the uncertainties acknowledged in the current manuscript

**(a) Justifying the contributions**

This study contributes to the field of research substantially by being the first to quantify AR driven inland compound flooding under climate change using large ensembles. To date, only two studies have explicitly addressed compound effects: one focusing on coastal flooding along the Dutch coast (Ridder et al., 2018), and the other on temporal compounding of atmospheric rivers in California (Bowers et al., 2024). Further, to our knowledge, no prior research has investigated inland compound flooding driven by atmospheric rivers, nor examined the combined influence of externally forced climate change and internal climate variability arising from natural variability of climate and large-scale climate patterns.

**(b) Clarification of uncertainties**

The uncertainties identified during this study are inherent to the problem being studied and we explicitly acknowledge them to highlight future research opportunities. Fully resolving the range of uncertainties identified would require separate in depth studies because of their complexity. The current body of literature highlights the extent of differences caused by the existence of different atmospheric river detection techniques (Ralph et al., 2019; Rutz et al., 2019). Internal climate variability is a known irreducible source of error, and numerous studies have examined

its influence on modeling various phenomena. This study specifically quantifies and discusses the implication of uncertainty from internal variability as a key objective. The results in the submitted work can support future research specifically targeting the relationship between internal variability and AR-driven activity, an important direction underscored by the strong link we identified between compound events and AR-related flooding.

In response to the comments on ensemble size, although a large ensemble may increase the accuracy of the results, the ensemble size used in this work aligns with prior large scale studies and yields consistent results (Hagos et al., 2016; Michaelis et al., 2022; Tseng et al., 2022).

**References**

Bowers, C., Serafin, K. A., and Baker, J. W.: Temporal compounding increases economic impacts of atmospheric rivers in California, Sci. Adv., 10, eadi7905, https://doi.org/10.1126/sciadv.adi7905, 2024.

Chen, X., Leung, L. R., Wigmosta, M., and Richmond, M.: Impact of Atmospheric Rivers on Surface Hydrological Processes in Western U.S. Watersheds, J. Geophys. Res. Atmospheres, 124, 8896–8916, https://doi.org/10.1029/2019JD030468, 2019.

Dettinger, M.: Historical and Future Relations Between Large Storms and Droughts in California, San Franc. Estuary Watershed Sci., 14, https://doi.org/10.15447/sfews.2016v14iss2art1, 2016.

Fereshtehpour, M., Najafi, M. R., and Cannon, A. J.: Characterizing Compound Inland Flooding Mechanisms and Risks in North America Under Climate Change, Earths Future, 13, e2024EF005353, https://doi.org/10.1029/2024EF005353, 2025.

Gillett, N. P., Cannon, A. J., Malinina, E., Schnorbus, M., Anslow, F., Sun, Q., Kirchmeier-Young, M., Zwiers, F., Seiler, C., Zhang, X., Flato, G., Wan, H., Li, G., and Castellan, A.: Human influence on the 2021 British Columbia floods, Weather Clim. Extrem., 36, 100441, https://doi.org/10.1016/j.wace.2022.100441, 2022.

Guan, B., Waliser, D. E., and Ralph, F. M.: Global Application of the Atmospheric River Scale, J. Geophys. Res. Atmospheres, 128, e2022JD037180, https://doi.org/10.1029/2022JD037180, 2023.

Hagos, S. M., Leung, L. R., Yoon, J.-H., Lu, J., and Gao, Y.: A projection of changes in landfalling atmospheric river frequency and extreme precipitation over western North America from the Large Ensemble CESM simulations, Geophys. Res. Lett., 43, 1357–1363, https://doi.org/10.1002/2015GL067392, 2016.

Lamjiri, M. A., Dettinger, M. D., Ralph, F. M., Oakley, N. S., and Rutz, J. J.: Hourly analyses of the large storms and atmospheric rivers that provide most of California's precipitation in only 10

- to 100 hours per year, San Franc. Estuary Watershed Sci., 16, 1–17, https://doi.org/10.15447/sfews.2018v16iss4art1, 2018.
- Michaelis, A. C., Gershunov, A., Weyant, A., Fish, M. A., Shulgina, T., and Ralph, F. M.: Atmospheric River Precipitation Enhanced by Climate Change: A Case Study of the Storm That Contributed to California's Oroville Dam Crisis, Earths Future, 10, e2021EF002537, https://doi.org/10.1029/2021EF002537, 2022.
- Mishra, A. N., Maraun, D., Schiemann, R., Hodges, K., Zappa, G., and Ossó, A.: Long-lasting intense cut-off lows to become more frequent in the Northern Hemisphere, Commun. Earth Environ., 6, 115, https://doi.org/10.1038/s43247-025-02078-7, 2025.
- Oakley, N. S., Lancaster, J. T., Kaplan, M. L., and Ralph, F. M.: Synoptic conditions associated with cool season post-fire debris flows in the Transverse Ranges of southern California, Nat. Hazards, 88, 327–354, https://doi.org/10.1007/s11069-017-2867-6, 2017.
- Pryor, S. C., McKendry, I. G., and Steyn, D. G.: Synoptic-Scale Meteorological Variability and Surface Ozone Concentrations in Vancouver, British Columbia, 1995.
- Radić, V., Cannon, A. J., Menounos, B., and Gi, N.: Future changes in autumn atmospheric river events in British Columbia, Canada, as projected by CMIP5 global climate models, J. Geophys. Res. Atmospheres, 120, 9279–9302, https://doi.org/10.1002/2015JD023279, 2015.
- Ralph, F. M., Neiman, P. J., Wick, G. A., Gutman, S. I., Dettinger, M. D., Cayan, D. R., and White, A. B.: Flooding on California's Russian River: Role of atmospheric rivers, Geophys. Res. Lett., 33, https://doi.org/10.1029/2006GL026689, 2006.
- Ralph, F. M., Wilson, A. M., Shulgina, T., Kawzenuk, B., Sellars, S., Rutz, J. J., Lamjiri, M. A., Barnes, E. A., Gershunov, A., Guan, B., Nardi, K. M., Osborne, T., and Wick, G. A.: ARTMIPearly start comparison of atmospheric river detection tools: how many atmospheric rivers hit northern California's Russian River watershed?, Clim. Dyn., 52, 4973–4994, https://doi.org/10.1007/s00382-018-4427-5, 2019.
- Ralph, F. M., Dettinger, M. D., Jonathan, R. J., and Waliser, D. E. (Eds.): Atmospheric Rivers, 1st ed., Springer International Publishing, 284 pp., 2020.
- Ridder, N., De Vries, H., and Drijfhout, S.: The role of atmospheric rivers in compound events consisting of heavy precipitation and high storm surges along the Dutch coast, Nat. Hazards Earth Syst. Sci., 18, 3311–3326, https://doi.org/10.5194/nhess-18-3311-2018, 2018.
- Rutz, J. J., Steenburgh, W. J., and Ralph, F. M.: Climatological Characteristics of Atmospheric Rivers and Their Inland Penetration over the Western United States, Mon. Weather Rev., 142, 905–921, https://doi.org/10.1175/MWR-D-13-00168.1, 2014.
- Rutz, J. J., Shields, C. A., Lora, J. M., Payne, A. E., Guan, B., Ullrich, P., O'Brien, T., Leung, L. R., Ralph, F. M., Wehner, M., Brands, S., Collow, A., Goldenson, N., Gorodetskaya, I., Griffith, H., Kashinath, K., Kawzenuk, B., Krishnan, H., Kurlin, V., Lavers, D., Magnusdottir, G., Mahoney, K., McClenny, E., Muszynski, G., Nguyen, P. D., Prabhat, Mr., Qian, Y., Ramos, A.

M., Sarangi, C., Sellars, S., Shulgina, T., Tome, R., Waliser, D., Walton, D., Wick, G., Wilson, A. M., and Viale, M.: The Atmospheric River Tracking Method Intercomparison Project (ARTMIP): Quantifying Uncertainties in Atmospheric River Climatology, J. Geophys. Res. Atmospheres, 124, 13777–13802, https://doi.org/10.1029/2019JD030936, 2019.

Sharma, A. R. and Déry, S. J.: Contribution of Atmospheric Rivers to Annual, Seasonal, and Extreme Precipitation Across British Columbia and Southeastern Alaska, J. Geophys. Res. Atmospheres, 125, e2019JD031823, https://doi.org/10.1029/2019JD031823, 2020a.

Sharma, A. R. and Déry, S. J.: Linking Atmospheric Rivers to Annual and Extreme River Runoff in British Columbia and Southeastern Alaska, J. Hydrometeorol., 21, 2457–2472, https://doi.org/10.1175/JHM-D-19-0281.1, 2020b.

Tseng, K.-C., Johnson, N. C., Kapnick, S. B., Cooke, W., Delworth, T. L., Jia, L., Lu, F., McHugh, C., Murakami, H., Rosati, A. J., Wittenberg, A. T., Yang, X., Zeng, F., and Zhang, L.: When Will Humanity Notice Its Influence on Atmospheric Rivers?, J. Geophys. Res. Atmospheres, 127, e2021JD036044, https://doi.org/10.1029/2021JD036044, 2022.